# Genetic Structure and Diversity of the Yellowbelly Threadfin Bream *Nemipterus bathybius* in the Northern South China Sea

Mu-Rong Yi [1,2], Kui-Ching Hsu [1], Jin-Xi Wang [1], Bo Feng [1,2], Hung-Du Lin [3,*] and Yun-Rong Yan [1,2,4,5,*]

[1] College of Fisheries, Guangdong Ocean University, Zhanjiang 524088, China; murong_yi@outlook.com (M.-R.Y.); joekchsu@yahoo.com.tw (K.-C.H.); jinxiwang96@126.com (J.-X.W.); fengb@gdou.edu.cn (B.F.)

[2] Marine Resources Big Data Center of South China Sea, Southern Marine Science and Engineering Guangdong Laboratory, Zhanjiang 524013, China

[3] The Affiliated School of National Tainan First Senior High School, Tainan 701, Taiwan

[4] Guangdong Provincial Engineering and Technology Research Center of Far Sea Fisheries Management and Fishing of South China Sea, Guangdong Ocean University, Zhanjiang 524088, China

[5] Center of Marine Fisheries Information Technology, Shenzhen Institute of Guangdong Ocean University, Shenzhen 518035, China

[*] Correspondence: varicorhinus@hotmail.com (H.-D.L.); tuna_ps@126.com (Y.-R.Y.);
Tel.: +886-6-2097821 (H.-D.L.); +86-0759-2396129 (Y.-R.Y.)

**Abstract:** The genetic structure and demography of the yellowbelly threadfin bream, *Nemipterus bathybius*, in the northern South China Sea were examined using the mitochondrial DNA cytochrome b gene (1141 bp). High levels of haplotype and nucleotide diversities (0.98 and $5.26 \times 10^{-3}$, respectively) showed that all populations exhibited a high level of genetic diversity. Analysis of molecular variance (AMOVA), $F_{ST}$ statistics, and haplotype networks suggested the absence of significant genetic differentiation along the coast of the northern South China Sea. Although the results suggested that the lack of differentiation within the population structure of *N. bathybius* was shaped by ocean currents, our results also showed that the Qiongzhou Strait limited their migration between Beibu Gulf and the northern South China Sea. Neutrality tests and mismatch distributions indicated population expansion, but the Bayesian skyline plots and approximate Bayesian computation approaches suggested that the population sizes recently contracted. The diversification of multiple stocks, which were induced by two ocean current systems, contributed to these discordant results. Although these analyses of demographic history revealed no evidence for recent population bottlenecks, the population demography needs to be evaluated further.

**Keywords:** mitochondria; ocean currents; qiongzhou strait; demography; multiple stocks

## 1. Introduction

China is adjacent to the northwest Pacific Ocean. The length of the coastline of mainland China, from the mouth of the Yalu River on the China–Korea border in the north to the mouth of the Beilun River on the China–Vietnam border in the south, is more than 18,000 km (km). The marginal seas of China include the Yellow Sea (including the Bohai Gulf), the East China Sea (including the Taiwan Strait), and the South China Sea (including the Beibu Gulf). Many phylogeographic studies conducted along the coastline of China have proposed that (1) the sea-level fluctuations of marginal seas were greatly impacted by Pliocene and Pleistocene glacial cycles [1–3], (2) the oceanographic current systems shape the migrations of marine species [4–7], and (3) the diluted water of the Yangtze River shapes coastal species differentiation by creating a phylogeographic break [5,8,9].

The macrogeographic population genetic structure among marine species in the Chinese seas has been intensively studied [9–14]. These populations of marine species are often characterized by low genetic divergence. Many studies have found that no significant genetic differentiation exists among populations of marine species along the

China coastline (e.g., *Periophthalmus modestus*, shuttles hoppfish [7], *Meretrix petechialis*, Asian hard clam [15], and groupers [16]). However, many studies found that on a small scale (<100 km), some populations of coastal species are genetically differentiated [17,18]. Peng et al. [19] found that two intertidal species, *Tetraclita japonica* and *Septifer virgatus*, were codistributed on small islands along the coast of Zhejiang, and these populations could be divided into two main metapopulations, exhibiting a northern and southern distribution. In addition, Kobayashi [20] found a clear intra-island population genetic structure of the sand bubbler crab *Scopimera ryukyuensis* within the Ryukyu Islands, e.g., at only 20 km. These results suggest that microgeographic genetic structure might exist among some marine species. Moreover, Liu [21] proposed that among the marginal seas of China, the South China Sea has the highest biodiversity. The bottom topography of the South China Sea is very complex, and it contributes to its high biodiversity. During the last glacial maximum, the Yellow and East China Seas were completely exposed, and the South China Sea became a semienclosed sea [22]. Thus, our study on the population genetic structure has focused on the northern South China Sea.

The yellowbelly threadfin bream, *Nemipterus bathybius* Snyder, 1911 (Perciformes, Nemipteridae), is a benthic fish widely distributed in the tropical and subtropical west Pacific region, ranging from southern Japan to Indonesia and northwestern Australia [23]. In China, the species is distributed in the coastal areas of the East China Sea and the South China Sea. *Nemipterus bathybius* is a valuable longline and bottom trawl fishery resource in the South China Sea [24,25]. The stocks of *N. bathybius* in the South China Sea have been severely overexploited since the 1990s [26]. The focuses of the previous studies of *N. bathybius* were on their age and growth parameters [25], fishery biology [26], and stock availability [27]. In contrast, there have been few studies involving molecular research (complete mitochondrial genome [28] and microsatellite markers [29]), and little is known about the molecular phylogeography and demographic history of *N. bathybius*. The species is most commonly found on sand or mud bottoms and is abundant at depths between 45 and 90 m, but larger fish generally occur at depths greater than 110–300 m [30]. The larger, older fish feed on crustaceans, fish, and cephalopods, and the younger, smaller fish feed on copepods, ostracods, and amphipods [30].

Accordingly, our study examined the population genetic structure and diversity of *N. bathybius* in the northern South China Sea using the genetic variability of the mitochondrial DNA (mtDNA) cytochrome *b* gene (cyt *b*), with the aim of revealing its demography. The sequences of mtDNA cyt *b* are usually used to reconstruct demographic history and animal phylogeography (e.g., *Garra orientalis* [31], *Tanichthys albonubes* [32], *Neocaridina* [33], *Scatophagus argus* [34], *Capra pyrenacia pyrenacia* [35]).

## 2. Materials and Methods

### 2.1. Sample Collection and DNA Sequencing

A total of 147 specimens of *N. bathybius* were collected from six localities across the northern South China Sea and Hainan Island in 2018 (Figure 1 and Table 1). For our study, specimens were collected from winter to spring (Table 1). All animal experiments were carried out in accordance with the guidelines and approval of the Animal Research and Ethics Committee of the College of Fisheries, Guangdong Ocean University (permission, B20200616-01). These localities were classified into three groups based on their locations. Four sampling localities were established off the coast of mainland China: (1) Shantou, ST; (2) Shanwei, SW; (3) Zhuhai, ZH; and (4) Yangjiang, YJ (Figure 1; Table 1). One sampling locality was established off Hainan Island: (5) Sanya, SY, and one locality was established off the Beibu Gulf: (6) Beibu Gulf, BBW.

The muscle tissue from the samples was fixed and stored in 95% ethanol. Genomic DNA was extracted using a Genomic DNA Purification Kit (Gentra Systems, Valencia, CA, USA). The complete mtDNA cyt *b* gene was amplified by polymerase chain reaction (PCR) using the primers JXcytb-F (5′-ATGGCTTGAAAAACCACCGT-3′) and JXcytb-R (5′-TCGGTTTACAAGACCGACGC-3′), which were designed based on the conserved

*Nemipterus* complete mitochondrial sequence region from NCBI. These two primers are located in the tRNA-Glu and tRNA-Thr genes, respectively. The PCR reactions were performed in 25 µL volumes. Each 25 µL PCR mixture contained 2.5 ng of template DNA, 2.5 µL of 10× reaction buffer, 2 µL of dNTP mix (10 mM), 5 pmol of each primer, and 2 U of Taq polymerase (TaKaRa, Taq polymerase). The PCR was conducted with an MJ Thermal Cycler as one cycle of denaturation at 94 °C for 3 min, 40 cycles of denaturation at 94 °C for 30 s, annealing at 53 °C for 45 s, and extension at 72 °C for 1 min 30 s, followed by a 72 °C extension for 10 min and 4 °C for storage. The purified PCR products were sequenced using an ABI 377 automated sequencer (Applied Biosystems, Foster City, CA, USA). The chromatograms were assessed using CHROMAS software (Technelysium), and the sequences were manually edited using BIOEDIT 6.0.7 [36]. All nucleotide sequences were deposited in GenBank under accession numbers MN848893-MN849039.

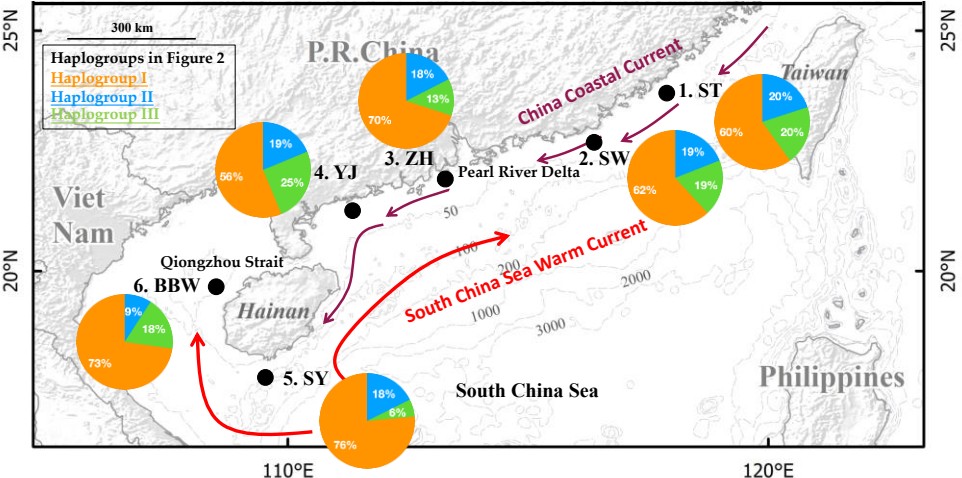

**Figure 1.** The sampling localities of *Nemipterus bathybius* are indicated by ●. The ocean currents are drawn based on Wang et al. [14]. The frequencies of the haplogroups (Figure 2) in each population are displayed on the map.

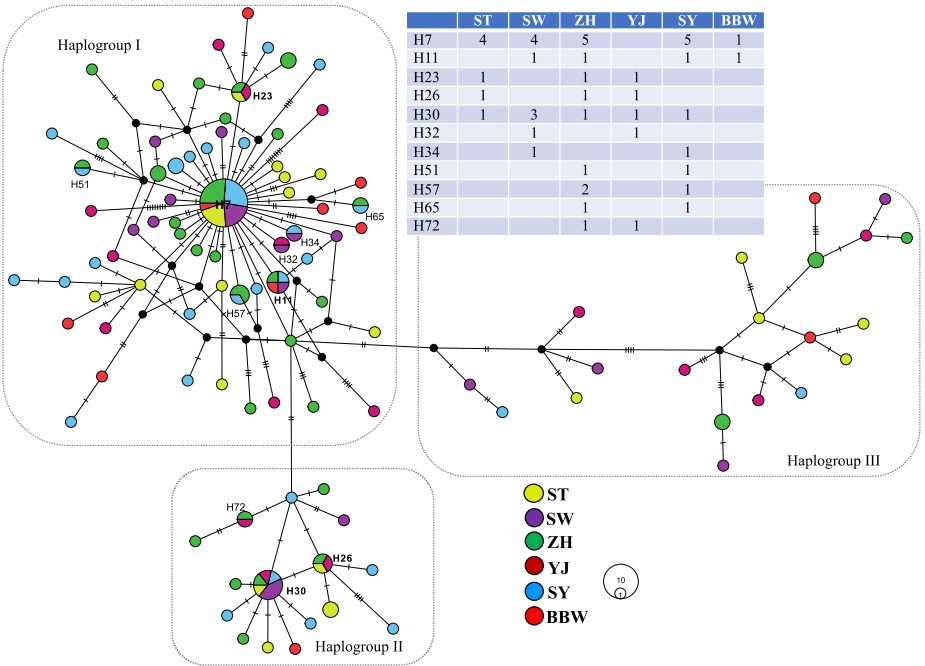

| | ST | SW | ZH | YJ | SY | BBW |
|---|---|---|---|---|---|---|
| H7 | 4 | 4 | 5 | | 5 | 1 |
| H11 | | 1 | 1 | | 1 | 1 |
| H23 | 1 | | 1 | 1 | | |
| H26 | 1 | | 1 | 1 | | |
| H30 | 1 | 3 | 1 | 1 | 1 | |
| H32 | | 1 | | 1 | | |
| H34 | | 1 | | | 1 | |
| H51 | | 1 | | | 1 | |
| H57 | | 2 | | | 1 | |
| H65 | | 1 | | | 1 | |
| H72 | | 1 | 1 | | | |

**Figure 2.** The network of *Nemipterus bathybius* in the South China Sea.

**Table 1.** The samples used for mtDNA analysis, location, code, and summary statistics in *Nemipterus bathybius*.

| Location | Code | Date | N | H | Hp | Hs | Hd | $\pi$ $(10^{-3})$ |
|---|---|---|---|---|---|---|---|---|
| 1. Shantou | ST | 15 April | 25 | 21 | 17 | 4 | 0.98 | 5.52 |
| 2. Shanwei | SW | 14 April | 21 | 16 | 11 | 5 | 0.96 | 4.62 |
| 3. Zhuhai | ZH | 11 April | 40 | 31 | 22 | 9 | 0.98 | 4.80 |
| 4. Yangjiang | YJ | 11 April | 16 | 16 | 11 | 5 | 1.00 | 8.00 |
| 5. Sanya | SY | 29 March | 34 | 29 | 22 | 7 | 0.98 | 4.36 |
| 6. Beibu Gulf | BBW | 1 January | 11 | 11 | 9 | 2 | 1.00 | 6.69 |
| Total | | | 147 | 103 | 92 | 11 | 0.98 | 5.26 |

N, number of specimens; H, number of haplotypes; Hp, number of private haplotypes; Hs, number of haplotypes shared among populations; Hd, haplotype diversity; $\pi$, nucleotide diversity.

### 2.2. Diversity and Structure Analyses

The sequences were aligned in the program CLUSTALX v1.81 [37] and then optimized manually. Levels of intrapopulation genetic diversity were measured by haplotype diversity (h) [38] and nucleotide diversity ($\pi$) [39] using DnaSP v5.0 software [40]. The haplotype list was detected by DnaSP. Best-fit models for the Bayesian analysis were inferred based on the Akaike Information Criterion (AIC) using PhyML with Smart Model Selection (http://www.atgc-montpellier.fr/phyml-sms/, accessed on 13 April 2021) [41]. GTR+G+I was selected as the most appropriate model for the subsequent analyses. The median-joining algorithm from Network 5.0 was used to reconstruct the haplotype networks [42]. The population clusters were estimated based on the $F_{CT}$ values using SAMOVA 1.0 [43]. As there were six sampling localities, the tests were performed with K-values of 2–5. The value for which $F_{CT}$ was the highest was chosen as the best grouping. Pairwise $F_{ST}$ values among the populations and the analyses of molecular variance (AMOVA) were performed with Arlequin v3.5 [44]. AMOVA partitioning was used to observe the within-population ($F_{ST}$), within-group ($F_{SC}$), and among-group ($F_{CT}$) components of the variation among the samples based on the groups defined in SAMOVA, the geographic locations, and the phylogenetic analyses. Pairwise uncorrected *p*-distances were estimated by MEGA-X [45].

### 2.3. Demographic Analyses

Historical demographic expansions were examined using four different approaches. First, Tajima's D [46] and Fu's *Fs* [47] tests were used in DnaSP. Second, demographic history was examined using mismatch distribution analyses implemented in DnaSP. Third, the effective population size changes over time were evaluated using Bayesian skyline plots (BSPs) in BEAST v1.8.0 [48]. A Bayesian skyline tree was selected, and a strict clock model was used. A mutation rate of 0.75–2.2% per million years (myr) has been calibrated for the mtDNA cyt *b* gene in multiple bony fishes for population expansion [49]. We ran the model for $3 \times 10^7$ generations to ensure the convergence of all parameters (ESSs > 200); the first 10% of the samples for each chain were discarded as burn-in. We used the same setting for two runs to check for convergence with $3 \times 10^7$ and sampling every $1 \times 10^3$ generations. Plots for each analysis were drawn using Tracer v1.6 [50].

Finally, an approximate Bayesian computation (ABC) framework was used to determine the population demography using the software DIYABC ver. 2.0 [51]. The reference table was built with 1,000,000 simulated data sets per scenario using all statistics. We used uniform priors for all scenarios and placed no constraints on the population sizes and coalescent times. The posterior probabilities were compared by logistic regression. To examine the past transition of the effective population size, our study first tested three simple population demographic scenarios (ABC 1 analyses). The scenarios were as follows: in scenario 1 (constant model), the effective population size was constant at N1 from the present to the past. In scenario 2 (bottleneck model), the effective population size changed from Na to N1 at t, and Na was larger than N1. In scenario 3 (expansion

model), the effective population size changed from Na to N1 at t, and Na was smaller than N1. However, the history of the population size change is not always simple, and the populations might suffer from isolation. Thus, following the observations in the results for ABC 1 and the other demographic analyses, our study proposed two scenarios in the ABC 2 analysis. These two scenarios were as follows: in scenario A (constant model), the effective population size was constant at N1 from the present to the past. In scenario B (isolation model), the populations diverged, and the effective population size changed at time t.

## 3. Results

### 3.1. Genetic Diversity

The complete cyt *b* gene comprises 1141 nucleotides, which consist of 16.1% guanine, 24.2% adenine, 30.4% thymine, and 29.3% cytosine (45.4% GC content). A total of 138 variable sites were observed, of which 52 were parsimony informative. A total of 103 haplotypes were identified among the 147 individuals (Table 1). Among the 103 haplotypes, 11 haplotypes were shared among more than two populations (Figure 2). The most widely distributed haplotype was H7, distributed in five populations (ST, SW, ZH, SY, and BBW). Population ZH had the most shared haplotypes, and population BBW included the fewest shared haplotypes. The average haplotype diversity was high (0.98), ranging from 1.00 (BBW and YJ) to 0.96 (SW), and the average nucleotide diversity was also high ($5.67 \times 10^{-3}$), ranging from $4.36 \times 10^{-3}$ (SY) to $8.00 \times 10^{-3}$ (YJ) (Table 1).

### 3.2. Population Differentiation

Median-joining network analysis showed that the haplotypes from all populations were mixed, although all haplotypes could be assorted as three haplogroups (I–III) (Figure 2). Each haplogroup contained all populations. Haplogroup I contained 99 specimens belonging to 67 haplotypes; haplogroup II contained 26 specimens belonging to 16 haplotypes; and haplogroup III contained 22 specimens belonging to 20 haplotypes. The highest frequencies of these three haplogroups were observed in populations SY (76%), ST (20%), and YJ (25%) (Figure 1). The mean *p*-distance among these three haplogroups was $0.826 \times 10^{-3}$, ranging from $0.542 \times 10^{-3}$ (between haplogroups I and II) to $1.016 \times 10^{-3}$ (between haplogroups II and II). The two fixation indices $N_{ST}$ and $G_{ST}$ were both zero. This result indicates an absence of phylogeographic structure [52]. Excluding the pairwise $F_{ST}$ between population SY and populations ST, ZH, and YJ, the values of the pairwise $F_{ST}$ were zero (Table 2). These results also indicated a lack of genetic differentiation among populations. The distribution of the shared haplotypes revealed that the population BBW and the other populations had fewer shared haplotypes, with a range from zero (population YJ) to two (populations ST, ZH, and SY); population ZH and the other populations had more shared haplotypes, with a range from two (population BBW) to six (population SY) (Table 2 and Figure 2).

The pairwise *p*-distance revealed that population YJ and the other populations had larger genetic distances (Table 2). In the SAMOVA analysis, the highest $F_{CT}$ value ($F_{CT} = 0.019$) occurred at K = 5 with a grouping arrangement of ST, YJ, SY, BBW, and SW + ZH (Table 3), but variability among groups was very rare (1.94%, Table 3). In addition, because the program SAMOVA could not work when n (number of populations) was equal to k (number of groups), our study calculated the *p*-distance among k groups (Table 3). When each locality corresponded to a genetic cluster (k = 6), the mean *p*-distance was not the highest. Although our study tested several schemes with AMOVA, the majority of the variability was accounted for within populations (99.36% to 100.04%) (Table 4). Based on these results, almost all variations did not exist among the sampling populations. These results suggest that the variations were distributed within each locality (Tables 3 and 4). Furthermore, our study revealed higher variations (55.37%) among the three haplogroups in the haplotype network (Figure 2 and Table 4).

**Table 2.** Matrix of pairwise $F_{ST}$ (below the diagonal), *p*-distance ($10^{-2}$, above the diagonal), and the number of shared haplotypes (Figure 2; underline, below the diagonal). Refer to Table 1 for the abbreviations of the localities. Values that are significantly different from zero (*p* = 0.05) are marked with an asterisk (*).

|  | ST | SW | ZH | YJ | SY | BBW |
|---|---|---|---|---|---|---|
| ST |  | 0.500 | 0.512 | 0.663 | 0.502 | 0.599 |
| SW | 0.0002 |  | 0.463 | 0.625 | 0.446 | 0.554 |
| ZH | 0.0004 | 0.0003 |  | 0.632 | 0.461 | 0.563 |
| YJ | 0.0003 | 0.0002 | 0.0004 |  | 0.629 | 0.721 |
| SY | 0.015 *2 | 0.0004 | 0.006 *6 | 0.077 *1 |  | 0.548 |
| BBW | 0.0001 | 0.0002 | 0.0002 | 0.0000 | 0.0002 |  |

**Table 3.** Analysis of the SAMOVA tests. See Table 1 for the locality codes. K, number of groups in the SAMOVA tests; %, variance among groups; D, mean pairwise *p*-distance.

| K | Groupings | $F_{CT}$ | % | D ($10^{-3}$) |
|---|---|---|---|---|
| 2 | (SY) (BBW,SW,ST,YJ,ZH) | 0.015593 | 1.56 | 0.500 |
| 3 | (SY) (YJ) (BBW,ST,SW,ZH) | 0.016767 | 1.68 | 0.585 |
| 4 | (ST) (SY) (YJ) (BBW,SW,ZH) | 0.0168116 | 1.68 | 0.572 |
| 5 | (ST) (SY) (YJ) (BBW) (SW,ZH) | 0.0194067 | 1.94 | 0.582 |
| 6 | (ST) (SY) (YJ) (BBW) (SW) (ZH) | - | - | 0.561 |

**Table 4.** Analysis of molecular variance (AMOVA) tests. See Table 1 for the locality codes.

| Scheme | Category Description | % Var. | Statistic | *p* |
|---|---|---|---|---|
| 1. Three geographic groups: (mainland) (Hainan Island) (Beibu Gulf) | | | | |
| | Among groups | 1.22 | $F_{SC} = 0.00$ | <0.578 |
| | Among populations in group | −0.93 | $F_{ST} = 0.00$ | <0.842 |
| | Within population | 99.71 | $F_{CT} = 0.01$ | <0.136 |
| 2. Five SAMOVA groups: (ST) (SY) (YJ) (BBW) (SW, ZH) | | | | |
| | Among groups | 1.94 | $F_{SC} = -0.02$ | <0.714 |
| | Among populations in groups | −1.98 | $F_{ST} = 0.00$ | <0.572 |
| | Within populations | 100.04 | $F_{CT} = 0.02$ | <0.065 |
| 3. Three NETWORK groups in Figure 2 | | | | |
| | Among groups | 55.37 | $F_{SC} = 0.02$ | <0.266 |
| | Among populations in group | 1.04 | $F_{ST} = 0.56$ | <0.001 |
| | Within populations | 43.59 | $F_{CT} = 0.55$ | <0.001 |

### 3.3. Demography

A signature of recent demographic expansion was detected with Tajima's D (−2.476, *p* < 0.01) and Fu's *Fs* (−152.268, *p* < 0.001) tests and the unimodal mismatch distribution fit the sudden expansion model well (*p* < 0·001) (Figure 3A). The BSP, which simulated the fluctuations in the population size over time, also revealed a population growth pattern and a relatively recent and extreme demographic expansion after a constant population size was found, which began approximately 0.07–0.25 myr ago (Figure 3B). However, a stable population size was found again recently, which began approximately 0.02–0.07 myr ago. Based on the Bayesian coalescent method, the most recent common ancestor ($T_{MRCA}$) of the total haplotypes dated back to approximately 0.34–0.98 (95% CI: 0.20–1.26) myr ago, which coincides with the late Quaternary epoch.

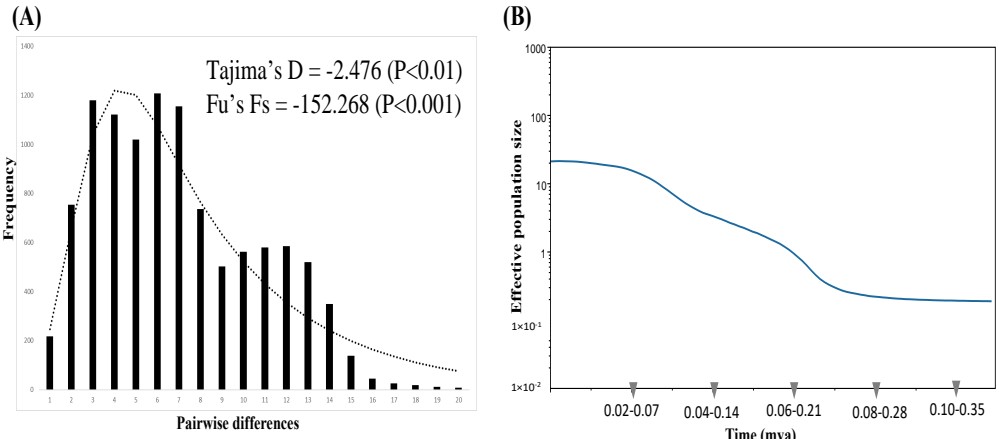

**Figure 3.** (**A**) The observed (bars) and expected (lines) mismatch distributions for all samples. (**B**) Bayesian skyline plot of all samples. Blue lines represent the median estimates of the effective population size, while the colored areas represent the upper and lower 95% highest posterior density limits. The y-axis is the product of the effective population size (Ne) and the generation length on a log scale, while the x-axis is the time scale before the present in units of a million years ago.

In the ABC1 analyses, three possible scenarios were designed, and the highest value of the posterior probability was observed for scenario 1 (0.9974, 95% CI: 0.9933–1.000), reflecting that the effective population size was constant from the present to the past (Figure 4). However, this result (Figure 4) and the results from the other methods (Figure 3) were inconsistent. Thus, our study proposed a second ABC analysis, ABC2. In ABC2, the highest value of the posterior probability was observed for scenario B (the isolation model; 1.000, 95% CI: 1.000–1.000). The results showed that the confidence intervals in scenarios 1 and B were unusually high, and the observed dataset did not fall within the cloud of simulated points in the pre-evaluation step of the scenarios tested (data not shown). These results might have resulted from less information (only 52 parsimony-informative sites). However, these results could also show that the populations of *N. bathybius* in the northern South China Sea were divergent.

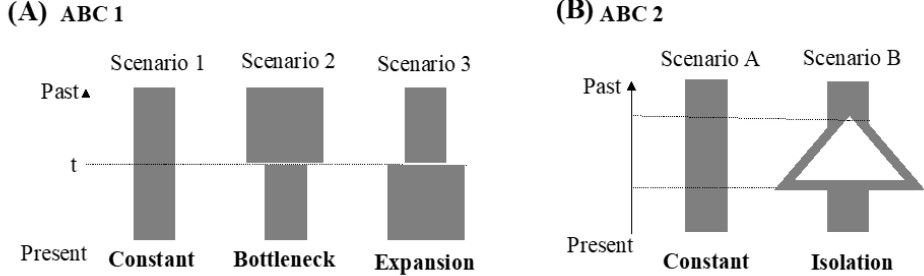

**Figure 4.** (**A**) Graphical representation of the three scenarios of demographic history used in the ABC1 analyses. (**B**) Graphical representation of the two scenarios of demographic history used in the ABC2 analyses.

## 4. Discussion

### 4.1. Genetic Structure and Diversity

The South China Sea connects with the East China Sea, Pacific Ocean, Sulu Sea, and Java Sea directly and connects with the Sea of Japan and the Indian Ocean indirectly through ocean currents. Oceanographic current systems, including the North and South China Coastal and Kuroshio Currents, shape the migration of marine and amphidromous species [6,15,53,54]. Previous studies have suggested that there were unlimited migrations between the South China Sea and the adjacent seas [7,16,55]. Within the South China Sea, previous studies also displayed unlimited migrations among populations [4,14,15,34,56].

In addition, Lim et al. [57] found that unlimited migrations existed among the populations of another bream species, *N. japonicus*, along the southern coasts of Peninsular Malaysia. Thus, unlimited migrations among populations of *N. bathybius* in the northern South China Sea are expected.

All of the results in the present study, including AMOVA, $F_{ST}$, and the haplotype network, suggested that *N. bathybius* in the northern South China Sea displayed a lack of differentiation within the population structure. Moreover, for our study, specimens were sampled in winter and spring (Table 1). In these seasons, the major monsoon currents were winter ocean current systems. The winter ocean current along the coast of China is the North China Coastal Current, which flows southward from the Yellow Sea and the East China Sea to the South China Sea (Figure 1) [14]. In addition, the South China Sea Warm Current flows northeastward over the continental slope and shelf in the northern South China Sea in all seasons [58]. Moreover, the results of the demographic analyses and the haplotype network suggested that the high diversity was contributed by multiple stocks (Figures 2–4). Thus, our study suggested that these ocean current systems brought different stocks to the northern South China Sea, where they became mixed.

Many researchers have discussed that the ocean current systems along the coastline of China shape the phylogeographic patterns of marine species [14,15,59]. However, the ocean current systems between the South China Sea and Beibu Gulf are rarely discussed. The Qiongzhou Strait, which separates the Beibu Gulf from the northern South China Sea, is generally shallower than 60 m. The restriction of migrations by the Qiongzhou Strait was supported by studies of the toothed top shell, *Monodonta labio* [60], and the small lacustrine goby, *Gobiopterus lacustris* [61]. Our study found that the pairwise *p*-distance showed the highest divergence between populations BBW and YJ, and the populations BBW and YJ did not have shared haplotypes (Figure 2 and Table 2). Thus, our study's results also suggested that the gene flow of *N. bathybius* from the northern South China Sea to the Beibu Gulf through the Qiongzhou Strait was interrupted.

In addition, among our sampled populations (excluding the population BBW), population YJ was located in the coastal area, and population SY was located in the offshore area (Figure 1). The number of shared haplotypes and pairwise *p*-distance revealed that population YJ was different from the other populations (Table 2). The genetic diversity was the highest in population YJ (Table 1). Thus, our study suggested that the population YJ might act as a sink. Moreover, there is a huge nuclear power plant near the sampling locality YJ. The water temperature might be higher there than in the other localities. *Nemipterus bathybius* is distributed at latitudes between 34°N and 23°S. In winter, the locality YJ may be a wintering ground.

In contrast, the results of pairwise $F_{ST}$ showed that genetic differences between population SY and the other populations were significant (Table 2). Thus, our study results supported the idea that population SY might act as a source to populate the seas in China. These results correspond to the biological characters of *N. bathybius*, with larger fish occurring in deeper waters [30]. The South China Sea Warm Current boosts the migration of *N. bathybius* to the SY coastal areas (YJ and BBW).

*4.2. Demographic History*

Many studies of marine species have revealed a population expansion before or after the Last Glacial Maximum (pre-LGM or post-LGM) [6,62,63]. Compared with previous studies in the same sea area [34,64], the results of the BSP for *N. bathybius*, *S. argus*, *T. japonicus*, and *T. nanhaiensis* all displayed population expansions during the post-LGM (Figure 3). However, the BSP also showed that the populations of *S. argus* and *T. nanhaiensis* encountered population bottlenecks, and *T. japonicus* and *N. bathybius* did not. *Trichiurus japonicus* is mainly distributed in the northern South China Sea, and it migrates southward seasonally; *Scatophagus argus* and *T. nanhaiensis* are mainly distributed in the southern South China Sea, while *N. bathybius* is widely distributed in the tropical and subtropical western Pacific region. During glaciation, East China was exposed completely, and the

South China Sea became a semi-enclosed sea. Our study considered that these different population histories shaped their different demographic patterns.

In this study of *N. bathybius*, the results of the neutral tests (Tajima's D and Fu's *F*s), mismatch distribution, and BSP all supported population expansion after the LGM (Figure 3). The results of the ABC analyses revealed that the population size was constant (Figure 4). However, the estimates of current ($\theta\pi = 5.26 \times 10^{-3}$) and historical ($\theta\omega = 2.284 \times 10^{-3}$) genetic diversity support the hypothesis that the population of *N. bathybius* shows a pattern of decline ($\theta\pi < \theta\omega$) [65]. Why did these analyses display incongruent results?

Our study reviewed the results of the mismatch distribution and BSP again (Figure 3). The mismatch distribution seems to be bimodal (Figure 3A), and the BSP also revealed a recent constant population size (Figure 3B). The bimodal mismatch distribution revealed high population divergences [34]. The results of the ABC2 analysis also supported the idea that the populations of *N. bathybius* were isolated (Figure 4B). Moreover, the network showed that all haplotypes could be assorted into three haplogroups, and the results of the AMOVA analyses displayed higher variations among these three haplogroups (Figure 2; Table 4). Compared with previous studies (e.g., *Trichiurus nanhaiensis*, *Cirrhimuraena chinensis*, *Saurida undosquamis*, *Scatophagus argus*) in the same sea area [34,64,66,67], the genetic diversity of *N. bathybius* was higher.

According to their biological characteristics, *N. bathybius* migrates over longer distances from deep-shallow areas, and the adults live far from coastal areas. The spawning ground of *N. bathybius* is not in a coastal environment. Several groups of *N. bathybius* foraging along the coast of China would be expected. In the description above (4.1. Genetic Structure and Diversity), our results suggest that the ocean current systems brought different stocks to the northern South China Sea, and then they mixed. The populations were isolated, and lineage sorting was due to higher divergence among the populations. Thus, our study identified higher divergences within *N. bathybius*, and diversification was contributed to by multiple stocks. The higher divergence resulted in neutral tests (Tajima's D and Fu's *F*s), and the mismatch distribution supported population expansion (Figure 3A). Actually, the BSP and ABC analyses showed that the population size was constant in recent years (Figures 3B and 4). Thus, the population demography of *N. bathybius* needs to be evaluated further.

## 5. Conclusions

Our study found that although the migrations of *N. bathybius* were shaped by ocean current systems, the migrations were also interrupted by the Qiongzhou Strait. The different ocean current systems brought different stocks to the northern South China Sea, and then they mixed. Our study also found that coastal populations act as sinks, and offshore populations act as sources in winter. Due to overexploitation and the deterioration of habitat environments over recent decades, the wild resources of *N. bathybius* have experienced a dramatic decline since the 1990s [26]. Although our results showed that the population size of *N. bathybius* was constant recently, beginning approximately 0.02–0.07 myr ago (Figures 3B and 4), we suggest that resource management and conservation of *N. bathybius* be considered. Our study proposes two policies. First, the coastal habitats should be protected. Second, an annual catch limit policy could be implemented in offshore fishing grounds. Although the sample sizes are limited, these results could help with future studies in biodiversity, fisheries, and conservation in the South China Sea.

**Author Contributions:** M.-R.Y.: Conceptualization, Methodology, Software, Formal analysis, Writing—original draft, Supervision, Writing—review and editing. K.-C.H.: Methodology, formal analysis, Writing—original draft, Validation, Writing—review and editing. J.-X.W.: Data curation, Investigation, Writing—review and editing. B.F.: Data curation, Investigation, Writing—review and editing. H.-D.L.: Conceptualization, Methodology, Visualization, Supervision, Writing—review and editing. Y.-R.Y.: project administration, conceptualization, validation, software, visualization, supervision, Writing—review and editing. All authors have read and agreed to the published version of the manuscript.

**Funding:** This study was supported by the National Key R & D Program of China (Grant Number: 2018YFD0900905), the Southern Marine Science and Engineering Guangdong Laboratory (Zhanjiang) (Grant Number: ZJW-2019-08), the National Natural Science Foundation of China (Grant Number: U20A2087), the Guangdong Basic and Applied Basic Research Foundation (Grant Number: 2019B1515120064), the Marine Economy Development Special Foundation of Department of Natural Resources of Guangdong Province (Grant Number: GDNRC[2020]052), and the Science and Technology Plan Projects of Guangdong Province, China, (Grant Number: 2018B030320006).

**Institutional Review Board Statement:** All animal experiments were carried out in accordance with the guidelines and approval of the Animal Research and Ethics Committee of College of Fisheries, Guangdong Ocean University (permissions, B20200616-01).

**Informed Consent Statement:** Not applicable.

**Data Availability Statement:** All nucleotide sequences were deposited in GenBank under accession numbers MN848893-MN849039.

**Acknowledgments:** We thank Po-Hsun Kuo, Department of Industrial Management, National Taiwan University of Science and Technology, Taipei, Taiwan, for helping with the data analysis.

**Conflicts of Interest:** The authors declare that they have no conflict of interest.

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
