# Peer review of "Genetic Structure and Diversity of the Yellowbelly Threadfin Bream Nemipterus bathybius in the Northern South China Sea"

_diversity, doi:10.3390/d13070324_

Round 1
Reviewer 1 Report
The research and methods presented in this paper are novel and interesting, but overall the paper needs improvement to meet the standards of the journal. Therefore, the authors need to reiterate and consider my queries. Also, GenBank sequence numbers need to be provided before final acceptance of the paper.
Line 50 I propose not to include the present finding. Save this thought for discussion.
Lines 64-65. differential habitat use as a function of fish age is a commonly observed pattern and depends mainly on food preferences that change along the ontogenetic development and the presence of ocean currents. But I don't see any real justification for microstructuring if this fish has pelagic eggs and a pelagic larval duration... Or are there some strong barriers preventing gene flow?
Lines 82-87. well, mtDNA is a good marker to reveal demographic population patterns and phylogeographic relationships. But for recent genetic structure and diversity, you can get a better picture with nuclear genes like SNPs or microsatellites. So I suggest the authors to focus more on demographic approach or include genes related to population studies.
Line 89. it is very difficult to make a conservation plan if some parameters are missing, such as current effective population size (nuclear), fixation index, bottleneck scenarios. As you mentioned, mtDNa is a very informative marker for demographic history analysis, but in this case, if you want to make a conservation plan, current population diversity, connectivity and structure are needed. Therefore, these expectations should be removed from the introduction
Line 102. what part of the tissue was fixed or the whole fish was fixed in ethanol? Please specify.
Line 109. reaction details must be available in the paper, in the main text or supplementary material!
Figure 1. please include a scale bar on the sampling card.
Line 194. network analysis is a good tool for interpreting the ancestral haplotypes found in the dataset, for showing connectivity between populations/regions, or for demographic interpretation (as an expansion with star-like shape). So, it cannot be put in the content of the population structure! Please, reword this paragraph.
Line 238. I found the confidence in the ABC2 - B scenario to be unusually high (1-1). Can the authors explain such a high confidence interval?
Line 246-247. clarification is needed here. It is not clear what "effective population size" stands for? Is it the mutation-scaled effective population size or the effective number of immigrants per generation. In the latter case, the mutation rate of loci per generation is not reported in M&M. MIGRATE program calculates the number of immigrants, so please be careful with the terminology used.
Lines 271-272. check the grammar here.
Discussion needs to be polished, there is a lot of repetition and many occasions where the sentence starts with the study results shows or similar. This way of writing Discussion needs to be corrected. The program MIGRATE is mentioned at least 15 times, which should be avoided.
Line 261-272. the repetition of unlimited migrations in every sentence should be avoided.
Lines 273-292. this paragraph should be written briefly, emphasizing the strong influence of flows on species migration patterns.
I recommend deleting subtitles in the discussion and creating a dynamic flow of discussion.
Lines 300-303. why do the authors link population diversity to population structure? Where is the connection?
Lines 310-317. repeat phase "these results might imply". Change that.
Lines 320-330. this is an opportunity to introduce the term barrier into the discussion!
Line 354-369. it is very difficult to follow this paragraph. The message is not clear and I get the impression that the authors are confused about the demographic history, current diversity and structure of the population.
Author Response
Open Review 1
The research and methods presented in this paper are novel and interesting, but overall the paper needs improvement to meet the standards of the journal. Therefore, the authors need to reiterate and consider my queries. Also, GenBank sequence numbers need to be provided before final acceptance of the paper.
Response: As requested, all results have been reviewed, and many sections including INTRODUCTION and DISCUSSION have been rewritten. All nucleotide sequences were deposited in GenBank under accession numbers MN848893-MN849039. We descripted in M&M.
Line 50 I propose not to include the present finding. Save this thought for discussion.
Response: As suggested, we move this sentence to discussion.
Lines 64-65. differential habitat use as a function of fish age is a commonly observed pattern and depends mainly on food preferences that change along the ontogenetic development and the presence of ocean currents. But I don't see any real justification for microstructuring if this fish has pelagic eggs and a pelagic larval duration... Or are there some strong barriers preventing gene flow?
Response: As requested, this section has been rewritten.
Lines 82-87. well, mtDNA is a good marker to reveal demographic population patterns and phylogeographic relationships. But for recent genetic structure and diversity, you can get a better picture with nuclear genes like SNPs or microsatellites. So I suggest the authors to focus more on demographic approach or include genes related to population studies.
Response: As requested, this section has been rewritten. Accordingly, our study examined the population genetic structure and diversity of N. bathybius in northern South China Sea using the genetic variability of the mitochondrial DNA (mtDNA) cytochrome b gene (cyt b), with the aim of revealing its demography and the migrations among sampling sites. The sequences of the mtDNA cyt b are usually used to reconstruct the demographic history and animal phylogeography.
Line 89. it is very difficult to make a conservation plan if some parameters are missing, such as current effective population size (nuclear), fixation index, bottleneck scenarios. As you mentioned, mtDNa is a very informative marker for demographic history analysis, but in this case, if you want to make a conservation plan, current population diversity, connectivity and structure are needed. Therefore, these expectations should be removed from the introduction
Response: As suggested, we deleted these expectations.
Line 102. what part of the tissue was fixed or the whole fish was fixed in ethanol? Please specify.
Response: The muscle tissue from samples were fixed and stored in 95% ethanol.
Line 109. reaction details must be available in the paper, in the main text or supplementary material!
Response: The PCR were performed in 25 μL volumes. Each 25 µl PCR reaction mixture contained 2.5 ng of template DNA, 2.5 µl of 10x reaction buffer, 2 µl of dNTP mix (10 mM), 5 pmol of each primer and 2U of Taq polymerase (TaKaRa, Taq polymerase). The PCR was programmed on an MJ Thermal Cycler as one cycle of denaturation at 94°C for 3 min, 40 cycles of denaturation at 94°C for 30 s, annealing at 53°C for 45 s and extension at 72°C for 1 min 30 s, followed by a 72°C extension for 10 min and 4°C for storage.
Figure 1. please include a scale bar on the sampling card.
Response: As requested, we added a bar for 300 km.
Line 194. network analysis is a good tool for interpreting the ancestral haplotypes found in the dataset, for showing connectivity between populations/regions, or for demographic interpretation (as an expansion with star-like shape). So, it cannot be put in the content of the population structure! Please, reword this paragraph.
Response: As requested, this section has been rewritten.
Line 238. I found the confidence in the ABC2 - B scenario to be unusually high (1-1). Can the authors explain such a high confidence interval?
Response: As requested, we do our best.
Line 246-247. clarification is needed here. It is not clear what "effective population size" stands for? Is it the mutation-scaled effective population size or the effective number of immigrants per generation. In the latter case, the mutation rate of loci per generation is not reported in M&M. MIGRATE program calculates the number of immigrants, so please be careful with the terminology used.
Response: As requested, this section has been rewritten. The effective number of migrants per generation was calculated for haploid data with female-transmission following the equation Nm = θ x M.
Lines 271-272. check the grammar here.
Response: As requested, this section has been rewritten.
Discussion needs to be polished, there is a lot of repetition and many occasions where the sentence starts with the study results shows or similar. This way of writing Discussion needs to be corrected. The program MIGRATE is mentioned at least 15 times, which should be avoided.
Response: As requested, this section has been rewritten.
Line 261-272. the repetition of unlimited migrations in every sentence should be avoided.
Response: As requested, this section has been rewritten.
Lines 273-292. this paragraph should be written briefly, emphasizing the strong influence of flows on species migration patterns.
Response: As requested, this section has been rewritten.
I recommend deleting subtitles in the discussion and creating a dynamic flow of discussion.
Response: As requested, we deleted some subtitles, and some sections has been rewritten.
Lines 300-303. why do the authors link population diversity to population structure? Where is the connection?
Response: This section should be connected to demography. As requested, this section has been rewritten.
Lines 310-317. repeat phase "these results might imply". Change that.
Response: As requested, we change it.
Lines 320-330. this is an opportunity to introduce the term barrier into the discussion!
Response: As requested, this section has been rewritten.
Line 354-369. it is very difficult to follow this paragraph. The message is not clear and I get the impression that the authors are confused about the demographic history, current diversity and structure of the population.
Response: As requested, all results have been reviewed, and many sections in DISCUSSION have been rewritten.

Reviewer 2 Report
This study targets the population structure of yellowbelly threadfin bream in the northern South China Sea based on one mitochondrial marker. In its current stage the ms is not ready for publication. in the following I list some specific comments:
1.line 42: "landforms of marginal seas"? What's that? I've never seen this term before. Maybe rephrase.
2. line 50: "our study"? Which study are you referring to here?
3. line 52: "intertidal species IN the small islands"????
4. line 56: "in only 20 km"? What do you mean here?
5. These were just a few examples. It continues like that throughout the mansucript. Proofreading by a native English speaker is required!
6. lines 58-68: Why do you expect and pop structure at all based on the biology of the species? From what you write here (habiat prefs etc.) I wouldn't expect any pop structure.
7. lines 68-71: The recent urban expansion is way too recent for leaving any signature in the pop structure (especially when you only look at one single mitochondrial maker).
8. line 84-86: also other genes are used for this pupose, and mtDNA is also used for other purpoes. Delete this sentence.
9. M&M, PCR: PCR details need to be given! Just writing that the details are available on request is not sufficient!
10: M&M, SAMOVA: You have 6 sampling sites, so you should test K-values up to 6!
11: M&M, pairwise distances in Mega: Why K2P distnaces? There's no justification for that. Either use uncorrected p-distances or the distance based on the best fitting substitution model (GTR+G+I in your case).
12: M&M, BEAST analysis: Why a rate of 1.05%? The rate you apply is from a cyprind and cyprinids are only very distantly related to sea breams. So there's no reason to assume that the subsitution rate is the same in these taxa. Better use a range (or min and max estimates) of substitution rates - quite a range of rates has been found in fishes. THe rates typically observed and applied for fish lie in the range of 0.75 - 2.2.% per MY (see e.g. Van Steenberge et al. 2020 J. Biogeogr., and refs therein).
13: M&M, DIYABC: You also need to give the info on the priors you employed, either in the main text or as Supplementary Material.
14: M&M, MIGRATE-N: Omit the MIGRATE-N analyses. Your results (network, FSTs) show that there's no pop structure, so no need for this analysis.
15: Results, Network: THis is an intersting network. It showas a lack of structure but also som divergent singletons, which likel respresent signatures of geneflow from outside your study area (see e.g. Sefc et al. 2020 J. Biogeogr.).
16: SAMOVA: There's no evidence for 4 clusters as i) you did not run the analyses up to K=6, and, more importantly, ii) SAMVOA cannot infer the support for K=1. And K=1 is what applies to your data (lack of structure - see network, ANOVA).
17: Results, Demography: Considere range of subsitution rates and also keep in mind the obvious time dependency of the molecular clock (Ho et al. 2005 Mbol. Biol. Evol.).
18: Table 2, FSTs: Are there any significant FSTs? Please also indicae the p-values.
19: Results, MIGRATE-N: Omit the MIGRATE-N-results and also the corresponding discussion. You have no structure, hence doing MIGRATE-N is entirely meaningless.
20: Discussion: no further commetns now, but you need to adapt it depending on the revised M&M and Results sections.
Author Response
Open Review 2
This study targets the population structure of yellowbelly threadfin bream in the northern South China Sea based on one mitochondrial marker. In its current stage the ms is not ready for publication. in the following I list some specific comments:
- line 42: "landforms of marginal seas"? What's that? I've never seen this term before. Maybe rephrase.
Response: As requested, this section has been rewritten.
- line 50: "our study"? Which study are you referring to here?
Response: As requested, this section has been rewritten.
- line 52: "intertidal species IN the small islands"????
Response: As requested, this section has been rewritten. Peng et al. [19] found that the two intertidal species, Tetraclita japonica and Septifer virgatus, were co-distributed in the small islands along the coast of Zhejiang, and these populations could be divided into two main metapopulations, exhibiting a northern and southern distribution.
- line 56: "in only 20 km"? What do you mean here?
Response: In marine, 20 km is a small scale (< 100 km) (Kobayashi 2020, Molecular Biology Reports 47: 2619-2626).
- These were just a few examples. It continues like that throughout the mansucript. Proofreading by a native English speaker is required!
Response: As requested, many sections including INTRODUCTION and DISCUSSION have been rewritten.
- lines 58-68: Why do you expect and pop structure at all based on the biology of the species? From what you write here (habiat prefs etc.) I wouldn't expect any pop structure.
Response: As requested, this section has been rewritten.
- lines 68-71: The recent urban expansion is way too recent for leaving any signature in the pop structure (especially when you only look at one single mitochondrial maker).
Response: As requested, this section has been rewritten.
- line 84-86: also other genes are used for this pupose, and mtDNA is also used for other purpoes. Delete this sentence.
Response: As requested, this section has been rewritten. We focus on demographic approach.
- M&M, PCR: PCR details need to be given! Just writing that the details are available on request is not sufficient!
Response: As requested, we added that. The PCR were performed in 25 μL volumes. Each 25 µl PCR reaction mixture contained 2.5 ng of template DNA, 2.5 µl of 10x reaction buffer, 2 µl of dNTP mix (10 mM), 5 pmol of each primer and 2U of Taq polymerase (TaKaRa, Taq polymerase). The PCR was programmed on an MJ Thermal Cycler as one cycle of denaturation at 94°C for 3 min, 40 cycles of denaturation at 94°C for 30 s, annealing at 53°C for 45 s and extension at 72°C for 1 min 30 s, followed by a 72°C extension for 10 min and 4°C for storage.
10: M&M, SAMOVA: You have 6 sampling sites, so you should test K-values up to 6!
Response: In SAMOVA analyses, our study had 6 sampling sites, the tests were performed with K-values of 2-5.
11: M&M, pairwise distances in Mega: Why K2P distnaces? There's no justification for that. Either use uncorrected p-distances or the distance based on the best fitting substitution model (GTR+G+I in your case).
Response: Most FST values were zero. We need to get more information. Thus, we estimate the pairwise genetic distance. As requested, we used uncorrected p-diatance.
12: M&M, BEAST analysis: Why a rate of 1.05%? The rate you apply is from a cyprind and cyprinids are only very distantly related to sea breams. So there's no reason to assume that the subsitution rate is the same in these taxa. Better use a range (or min and max estimates) of substitution rates - quite a range of rates has been found in fishes. THe rates typically observed and applied for fish lie in the range of 0.75 - 2.2.% per MY (see e.g. Van Steenberge et al. 2020 J. Biogeogr., and refs therein).
Response: The obvious time dependency of molecular clock is important for systematics. In demography, we just want to see the change of population sizes over time. A range of substitution rates did not provide more information. Moreover, the rate of 1.05% is the median of the range of 0.75-2.2%. Besides, Van Steenberge et al. (2020) focussed on freshwater fishes.
Van Steenberge MW, Vanhove MPM, Manda AC, Larmuseau MHD, Swart BL, Khang’Mate F, Arndt A, Hellemans B, Houdt JV, Micha JC, Koblmuller S, RoodtWilding R, Volckaert FAM. 2020. Unravelling the evolution of Africa’s drainage basins through a widespread freshwater fish, the African sharptooth catfish Clarias gariepinus. J. Biogeogr. 47(8), 1739-1754.
13: M&M, DIYABC: You also need to give the info on the priors you employed, either in the main text or as Supplementary Material.
Response: The reference table was built with 1,000,000 simulated data sets per scenario using all statistics. We used uniform priors for all scenarios and gave no constraints to population sizes and coalescent times. The posterior probabilities were compared by logistic regression.
14: M&M, MIGRATE-N: Omit the MIGRATE-N analyses. Your results (network, FSTs) show that there's no pop structure, so no need for this analysis.
Response: Our study did not use the MIGRATE-N analyses to examine the population structure. Our study used that to detect the gene flows between populations. Moreover, the results of the MIGRATE-N revealed much useful information.
15: Results, Network: THis is an intersting network. It showas a lack of structure but also som divergent singletons, which likel respresent signatures of geneflow from outside your study area (see e.g. Sefc et al. 2020 J. Biogeogr.).
Response: As requested, many sections have been rewritten.
16: SAMOVA: There's no evidence for 4 clusters as i) you did not run the analyses up to K=6, and, more importantly, ii) SAMVOA cannot infer the support for K=1. And K=1 is what applies to your data (lack of structure - see network, ANOVA).
Response: As requested, all results have been reviewed. However, when the sampling sites were 6, the K was 2-5. We did not run K=1 and K=6.
17: Results, Demography: Considere range of subsitution rates and also keep in mind the obvious time dependency of the molecular clock (Ho et al. 2005 Mbol. Biol. Evol.).
Response: The obvious time dependency of molecular clock is important for systematics. In demography, we just want to see the change of population sizes over time. A range of substitution rates did not provide more information.
18: Table 2, FSTs: Are there any significant FSTs? Please also indicae the p-values.
Response: As requested, we indicated the p-values.
19: Results, MIGRATE-N: Omit the MIGRATE-N-results and also the corresponding discussion. You have no structure, hence doing MIGRATE-N is entirely meaningless.
Response: Our study did not use the MIGRATE-N analyses to examine the population structure. Our study used that to detect the gene flows between populations. Moreover, the results of the MIGRATE-N revealed much useful information.
20: Discussion: no further commetns now, but you need to adapt it depending on the revised M&M and Results sections.
Response: As requested, many sections have been rewritten.

Round 2
Reviewer 1 Report
Dear authors,
I see that many comments and suggestions have been accepted, and the work as it stands is much improved. Nevertheless, I must make a few comments on the style of writing in the discussion. Every paragraph starts with Many studies or In this study/Our study. Please change this, there is no reason for such emphasis.
Regards
Author Response
Open Review 1
I see that many comments and suggestions have been accepted, and the work as it stands is much improved. Nevertheless, I must make a few comments on the style of writing in the discussion. Every paragraph starts with Many studies or In this study/Our study. Please change this, there is no reason for such emphasis.
Response: As requested, the manuscript was revised and edited for English language usage, grammar, spelling and punctuation by one or more native English-speaking editors on the AJE website using the verification code 3BE1-EE27-BFC7-5035-20FB.

Reviewer 2 Report
I've read the revision of this manusript, but I have to say there is only very little improvement. Most of my previous concerns are still valid. Apparently you did not fully understand them, especially when they concerned methodological issues. Hence, I try to be a bit bolder now with my comments:
- The language is still far from acceptable. Proofreading by a native English speaker from the ield or a professional editing service is reuqired (this is not just an option, I consider this obligatory). E.g., I'm pretty sure you didn't sample in the islands. Or, 20 km is definitely a short distance when it comes to marine organisms; I never doubted that, but the phrase itself is grammatically incorrect. These were some examples of language issues and NOT scietifical content.
- Abstract & thrughout the masnucript: Where is the evidence for winter migration? I can't find it (did I miss it somewhere)? In any case I would rephrase the title (an in particular omitting the term "winter migration").
- SAMOVA: You have 6 sampling sites. So you have a maximum of six geographic/genetic clusters (when each locality corresponds to a genetic cluster). Thus you need to test K = 2-6 and NOT only up to 5!
- molecular clock: It appears you completely misunderstood/misunderstand the molecular clock issue. So again. Why a rate of 1.05%? The rate you apply is from a cyprind and cyprinids are only very distantly related to sea breams. So there's no reason to assume that the subsitution rate is the same in these taxa. Better use a range (or min and max estimates) of substitution rates - quite a range of rates has been found in fishes. THe rates typically observed and applied for fish lie in the range of 0.75 - 2.2.% per MY (see e.g. Van Steenberge et al. 2020 J. Biogeogr., and refs therein). In your reply you mentioned that time dependency of the clock is only relevant for systmatics. This is simply wrong, as divergence times/expansion times or any recent demographic event inferred based on a rate obtained from deep phylogenetic splits will be severely affected by time dependency. In the meanwhile there's lots of publications on this issue. And you do need to consider a minimumand maximum subsitution rate for fish as otherwise your study will give the wrong impression of exactness, when there is in fact none. Yes, your 1.05% lie somewhere in the middle of the range typically observed in fish, but you do need to consider a plausible range in your results/discussion (to be conservative with the interpretation of the data). And this has nothing to do with freshwater versus marine fish. So, again, considering a plausible minimum/maximum of the range for fish is not just an option, but I consider this obligatory!
- I don't see any effect of the Pearl River on you pop genetic structure.
- MIGRATE-N: My previous comment is still valid. Inferring migration rates only makes sense if you have some kind of popualtion structure. Based on FSTs you seem to have a panmictic population (only SY seems to be somewhat differentiated from the rest of the pops, except for SW, which is kind of difficult to understand how this could be the case). So, this means that most localities belong to a single population. And inferring migration rates within a popualtion doesn't make sense at all! Just becuase MIGRATE-N gives you some output, this doesn't mean that this is meaningful in the context of your study. So, please delete the entire MIGRATE-N parts. You don't need these for your story.
Author Response
Open Review 2
I've read the revision of this manusript, but I have to say there is only very little improvement. Most of my previous concerns are still valid. Apparently you did not fully understand them, especially when they concerned methodological issues. Hence, I try to be a bit bolder now with my comments:
- The language is still far from acceptable. Proofreading by a native English speaker from the ield or a professional editing service is reuqired (this is not just an option, I consider this obligatory). E.g., I'm pretty sure you didn't sample in the islands. Or, 20 km is definitely a short distance when it comes to marine organisms; I never doubted that, but the phrase itself is grammatically incorrect. These were some examples of language issues and NOT scietifical content.
Response: We agreed with all the comments. The manuscript was edited for English language usage, grammar, spelling and punctuation by one or more native English-speaking editors on the AJE website using the verification code 3BE1-EE27-BFC7-5035-20FB.
However, the comment about the SAMOVA dealt a heavy below to our works. In SAMOVA, n populations = K groups, it is IMPOSSIBLE.
We never read the manuscripts in n=k.
We test K = 6 NOT only up to 5, but the programs, not only in SAMOVA but also in AMOVA, COULD NOT WORK.
If you consider to test each locality as a genetic cluster obligatory, we proposed a substitute plan. We calculate the mean p-distance among groups (Table 3), and this section have been rewritten.
- Abstract & thrughout the masnucript: Where is the evidence for winter migration? I can't find it (did I miss it somewhere)? In any case I would rephrase the title (an in particular omitting the term "winter migration").
Response: We sampled in winter and spring. In this season, the major monsoon currents were winter ocean current systems. Thus, we said “winter migration”. However, we have deleted the analyses of migrations. The title and some sections have been rewritten.
- SAMOVA: You have 6 sampling sites. So you have a maximum of six geographic/genetic clusters (when each locality corresponds to a genetic cluster). Thus you need to test K = 2-6 and NOT only up to 5!
Response: We try to test K = 6, BUT THE PROGRAMS, NOT ONLY IN SAMOVA BUT ALSO IN AMOVA, COULD NOT WORK. However, we proposed a substitute plan. We calculate the mean p-distance among groups (Table 3). However, the localities have shared haplotypes (Figure 2), and the pairwise FST values were almost “zero” (Table 2). Thus, the majority of the variability was not accounted for among localities. The sections have been rewritten.
- molecular clock: It appears you completely misunderstood/misunderstand the molecular clock issue. So again. Why a rate of 1.05%? The rate you apply is from a cyprind and cyprinids are only very distantly related to sea breams. So there's no reason to assume that the subsitution rate is the same in these taxa. Better use a range (or min and max estimates) of substitution rates - quite a range of rates has been found in fishes. THe rates typically observed and applied for fish lie in the range of 0.75 - 2.2.% per MY (see e.g. Van Steenberge et al. 2020 J. Biogeogr., and refs therein). In your reply you mentioned that time dependency of the clock is only relevant for systmatics. This is simply wrong, as divergence times/expansion times or any recent demographic event inferred based on a rate obtained from deep phylogenetic splits will be severely affected by time dependency. In the meanwhile there's lots of publications on this issue. And you do need to consider a minimumand maximum subsitution rate for fish as otherwise your study will give the wrong impression of exactness, when there is in fact none. Yes, your 1.05% lie somewhere in the middle of the range typically observed in fish, but you do need to consider a plausible range in your results/discussion (to be conservative with the interpretation of the data). And this has nothing to do with freshwater versus marine fish. So, again, considering a plausible minimum/maximum of the range for fish is not just an option, but I consider this obligatory!
Response: As requested, we used the range of 0.75 – 2.2 % to re-calculate the evolutionary times.
- I don't see any effect of the Pearl River on you pop genetic structure.
Response: As requested, the effects of the Pear River on the population genetic structure of have been deleted, and some sections have been rewritten.
- MIGRATE-N: My previous comment is still valid. Inferring migration rates only makes sense if you have some kind of popualtion structure. Based on FSTs you seem to have a panmictic population (only SY seems to be somewhat differentiated from the rest of the pops, except for SW, which is kind of difficult to understand how this could be the case). So, this means that most localities belong to a single population. And inferring migration rates within a popualtion doesn't make sense at all! Just becuase MIGRATE-N gives you some output, this doesn't mean that this is meaningful in the context of your study. So, please delete the entire MIGRATE-N parts. You don't need these for your story.
Response: As requested, we delete the entire MIGRATE-N parts, and some sections have been rewritten.

This manuscript is a resubmission of an earlier submission. The following is a list of the peer review reports and author responses from that submission.